# How Did the COVID-19 Pandemic Increase Salience of Intimate Partner Violence on the Policy Agenda?

**DOI:** 10.3390/ijerph20054461

**Published:** 2023-03-02

**Authors:** Luce Lebrun, Aline Thiry, Catherine Fallon

**Affiliations:** Department of Political Science, Faculty of Law, Political Science and Criminology, University of Liège, 4000 Liège, Belgium

**Keywords:** intimate partner violence, COVID-19, agenda-setting, Belgium, violence against women, lockdown, pandemic, domestic violence, policy window, multiple streams model, policy entrepreneurs

## Abstract

Belgian authorities, like most authorities in European countries, resorted to unprecedented measures in response to the spread of the COVID-19 pandemic between March 2020 and May 2022. This exceptional context highlighted the issue of intimate partner violence (IPV) in an unprecedented way. At a time when many other issues are being put on hold, IPV is being brought to the fore. This article investigated the processes that have led to increasing political attention to domestic violence in Belgium. To this end, a media analysis and a series of semi-structured interviews were conducted. The materials, collected and analyzed by mobilizing the framework of Kingdon’s streams theory, allowed us to present the agenda-setting process in its complexity and the COVID-19 as a policy window. The main policy entrepreneurs were NGOs and French-speaking feminist women politicians. Together, they rapidly mobilized sufficient resources to implement public intervention that had already been proposed in the preceding years, but which had been waiting for funding. By doing so, they responded during the peak of the pandemic to requests and needs that had already been expressed in a “non-crisis” context.

## 1. Introduction

On 11 March 2020, the World Health Organization recognized the COVID-19 epidemic as a pandemic and governments were called upon to take the necessary and unprecedented health measures, such as lockdowns, social distancing and “voluntary self-isolation”. While individual and collective vulnerabilities were strongly impacted, and women presented higher forms of stress [1,2,3,4], the specific issue of intimate partner violence caught international attention. Indeed, the pandemic context and the imposed lockdowns not only aggravated the factors contributing to domestic violence but also exacerbated the very effects of this abuse [5,6,7,8]. The psychological domestic violence was also noted as more prevalent than physical domestic violence [9,10]. On 5 April 2020, the UN Secretary-General António Guterres called for a ‘ceasefire’ in homes, as violence against women and girls surged (The Independent). Moreover, access to support also became a major issue [11]. Indeed, the lockdown of the population considerably limited the possibilities of assistance, whether in the form of professional help or help in the intimate sphere [12]. The health measures imposed also had an impact on the practices and cooperation processes between the different sectors involved in dealing with such violence. The exceptional context of the pandemic highlighted the issue of intimate partner violence in an unprecedented way. It seems to be making its way onto the political and media agenda. At a time when many other issues are being put on hold, violence against women, and more particularly violence committed in the private sphere, is being brought to the fore.

This phenomenon was also observed in Belgium and more particularly in the French-speaking part of the country which this paper focuses on. A “Task Force” was in fact created at the very beginning of the pandemic: “Task Force Violences conjugales et intrafamiliale.” Throughout this paper, we looked at the process that has led to this increasing political attention for domestic violence. Before we go further, we will in the following paragraphs say a few words about the Belgian context. 

### 1.1. Belgian Context

#### 1.1.1. The COVID-19 Pandemic in Belgium

From 18 March 2020 to May 2022, the Belgian authorities, like many neighboring countries, resorted to unprecedented measures in response to the spread of the COVID-19 pandemic. Strict population lockdowns were imposed to deal with the different waves of contamination of the virus: limitations on social contacts and mobility; a ban on gatherings; and the closure of schools etc. Other than the ‘waves’ of infection [13], there was no other sequencing for these two years. To provide a better understanding for potential readers who are not familiar with the context of this article, we will briefly review the measures that defined this period. On 18 March 2020, the Belgian population had to adhere to a strict lockdown, which lasted until the following 5 May. Only so-called essential travel was allowed. Only pharmacies, food stores and bookshops remained open. This was the first distinctly identifiable moment in Belgium since the start of the pandemic. A second key moment was the period between May 2020 and September 2020. During these months, a deconfinement plan was launched with a view to a return to normality. Phase 2 of this transitional period was marked by the reopening of schools on 18 May 2020. This was a moment that is worth highlighting in the context of this contribution, given the feedback from families with children. Couples with children, in contrast to those without children, were said to have experienced an increase in intimate partner violence in the early days of the pandemic [14]. In October 2020, new measures were adopted to deal with the resurgence of the pandemic. As the pandemic evolved, periods of lock-down and opening alternated. In March 2022, most of the sanitary measures were finally lifted. May 2022 marked the end of all measures. 

#### 1.1.2. Defining IPV and Related Policies in Belgium

Intimate Partner Violence (IPV) has gradually moved from the private sphere and been exposed to public interventions since the 1970s. IPV has been the subject of considerable conceptualization in recent decades. However, defining and naming this violence as a policy problem remains an issue. There is no universally accepted definition of such violence, while various theoretical approaches and explanations “offer sometimes fundamentally contradictory, even incompatible visions” [15]. Considering this issue and in the context in which this contribution is made, it is necessary to consider, in a non-exhaustive way, some elements of the specifically definition of IPV in the Belgian context.

In the wake of the feminist movements that brought the issue of violence against women to the forefront of public debate as a societal problem and no longer a private matter, the status of violence suffered within the home evolved. This process, which began in the 1970s, led to a move away from the invisibility and arbitrariness of the private sphere to become the subject of concerted public policy initiatives at the end of the century in many Western democracies [16]. In Belgium, a milestone was the change of law in 1997 which aimed at reducing domestic violence and “reinforces the protection of victims […]. Henceforth, the Public Prosecutor might enter the marital home or any other place based on a call or complaint from the victim” [17]. The year 2006 was also significant in the field of IPV, because the Federal Parliament adopted a resolution on 23 November which defined the fight of violence against women as a political priority and adopted a common definition of IPV, which has since become the reference point for public authorities in the sector. This was translated within the judiciary /police sectors under the COL3/2006 which defines “violence in intimate relationships [as] the manifestation, in the private sphere, of the unequal power relationship between women and men still at work in our society.” [18] It changed the frame of reference for violence in couples by placing it in the field of structural inequalities between women and men [15]. IPV was then defined for the first time at the Belgian federal level and criminal policy guidelines are drawn up [19]. Such a definition also makes it possible to break with the “previous focus on physical violence alone [and with the tacit tolerance] denounced ten years earlier when the Lizin (the name of the senator who brought the 1997 legislation aimed at combating violence within couple) Bill was introduced“ [19]. These juridical tools clearly contribute to the juridical definition of domestic violence in Belgium and are “essential for clarifying any ambiguities that might remain in the absence of a definition” [15]. 

The Council of Europe Convention on Preventing and Combating Violence against Women and Domestic Violence further contributed to defining domestic violence in Belgium. Signed in 2012 and ratified in 2016 by all parliaments in Belgium, the so-called Istanbul Convention recognizes that such violence is “a manifestation of the historically unequal power relations between women and men that have led to the domination and discrimination of women by men, thus depriving women of their full emancipation.” By adopting such a tool, authorities are supposed to give more attention to a gender-based approach and strategy for action. Indeed, the Convention “recognizes that women and girls are at a higher risk of gender-based violence than men. It notes that partner violence and other forms of gender-based violence disproportionately affect women while emphasizing that men can also be victims” (NAP 2015–2019) [20]. This is a legally binding instrument for signatory countries. An independent and specialized body, Grevio, regularly evaluates the level of engagement of each country (Grevio, 2019). The commitment made by ratifying this convention marks a major evolution in the conception of domestic violence in Belgium since it now serves as a “compass” (NAP 2021–2025) [21]. in the construction of public policies. The latest Grevio report in Belgium (2019) evaluated the public policies developed in the country to combat violence against women and domestic violence. It urged the authorities to address problems and shortcomings such as: the lack of coordination between the different entities of the country (the regional and federal authorities), the need to apply a gender perspective, and the need to grant more resources to the issue [22]. 

After the 2019 elections, three female ministers were in charge for gender equality in the newly elected French speaking governments. As feminists, they share a political will but also the desire to work together so that “women’s rights no longer suffer from scattering of competences [23].” They chose to develop as much cooperation as possible on gender-related issues. The occurrence of the pandemic and the lockdown in March 2020 prompted an increase in cooperation: they installed a specific task force aimed at tackling family violence, associating ministerial cabinets with public servants and NGOs. 

A review of these events provides a necessary introduction to the context in which this article is written. 

### 1.2. Aim of the Study and Concepts

In this paper, we asked if the COVID-19 pandemic has put the issue of intimate partner violence on the political agenda. To answer this question, the model of streams for agenda-setting developed by Kingdon [24] will serve as a key to “understanding why certain issues become more important on the agendas of government authorities while others disappear”. Indeed, it has already been highlighted that Kingdon’s model is relevant to address the pandemic as a significant policy window [25] creating opportunities for policy agenda reforms. Policy analysts such as Kingdon [24] note that problems move up and down the governmental agendas, often independently of the objective state of a problem [26]. Agenda-setting researchers assume that the governmental agenda (as well as media attention) can only address a limited number of issues at the same time: this limitation creates a competition among issues [26]. Kingdon’s [24] streams model of agenda-setting takes into consideration the dynamics of this competition and the interplay of three streams or policy processes: problems, policies, and politics. Windows of opportunity can open when a solution is attached to what policy actors perceive as a public problem. What Pralle/Kingdon call “policy entrepreneurs” must then seize the opportunity and push for government action [26]. In line with this model of analysis, we proposed to question, first, the reasons that allow one issue to become central to the detriment of others. Second, we analyzed the way in which politicians define their priorities and become policy entrepreneurs. Third, this paper highlighted the special role of the NGO sector during this period. By mobilizing its concepts, we were then able to define the main factors which led to this violence being placed on the political agenda during the pandemic. 

Before we progress further into this contribution, we will first detail the data collection. The latter was limited to the French-speaking part of Belgium only, la Fédération Wallonie Bruxelles. Our analysis was therefore specific to this territory. We will then present the results from the data collected. We will enter fully into the discussion of these results before we conclude. 

## 2. Materials and Methods 

The data collection on which this paper is based contributes to a contextualization of the impact of the COVID-19 pandemic on intimate partner violence. This forms part of an ongoing broader research project (B2/202/P3/IPV-DACOVID). To analyze the evolution of the media and policy agenda, data on the political-administrative framework and discourse analysis were collected. The data consisted of an analysis of the public debates that took place during the period of confinement and its aftermath, as well as an analysis of the press. Exploratory interviews enriched this collection. The following paragraphs will attempt to detail them. 

### 2.1. Media Analysis 

Media analysis considers, on the one hand, the ‘reported’ events, and on the other hand the ‘commented’ events that decipher reported information and thus contribute fully to the construction of a ‘problematized space’ [27]. We are interested in the way in which the issue of intimate partner violence was dealt with by the French-speaking Belgian media during the COVID-19 pandemic. We consider the media to be a fully-fledged player in the ‘political game’ [28]. Their importance in shaping the public agenda has long been recognized: ‘the press may not be successful much of the time in telling its readers what to think, but it is stunningly successful in telling its readers what to think about’ [29]. 

Several media were then selected for analysis and ‘to draw from this limited corpus lines of force and emerging themes’ [30]. To cover the French-speaking press, the following media were selected: Le Soir, La Libre, La DH, RTBF. These are the leading French-speaking daily newspapers in terms of readership and a public television. To gather as many publications as possible about intimate partner violence, the following keywords were used on newspaper search engines: intimate partner violence, intrafamily violence, domestic violence, violence against women, sexual violence, gender violence, feminicide. In concrete terms, this data collection allowed the construction of a continuously updated table containing articles, opinion pieces, press releases, etc. published since 17 March 2020 on the issue. Through this collection, an initial overview of the different events that have occurred since the beginning of the health crisis, their chronology, and the discourses on them has gradually emerged. The media analysis put at the fore a series of initiatives developed throughout the country to address the issue of an increase in IPV’s records.

### 2.2. Semi-Structured Interviews

Through the conduct of interviews (n = 17), and within the framework of an abductive approach, the purpose of exploratory interviews is not to verify hypotheses, but rather to become aware of dimensions and aspects of a problem that the researcher would probably not have thought of spontaneously [31]. The persons approached for these interviews were members of the civil society (Table 1) within the Fédération Wallonie Bruxelles as well as representatives of different ministerial cabinets and members of the administrations (Table 2), all French-speaking and participants in the “Task force violence conjugale et intrafamiliale” launched at the end of March 2020. 

Due to sanitary conditions, these interviews were all carried out remotely. Potential respondents were first selected through the media analysis, with some names coming up regularly and attracting our attention. The ‘snowball’ method was then used to identify the actors needed to cover the debates and cooperation processes in the Task Force. The semi-structured interviews were based on an interview grid, a framework [32] adapted according to the respondent and his sector of activity. The entire contents of the interviews were recorded for analysis. This interview grid can be found in the Table 3.

Taking an interpretive-analysis approach [33], we posited that beliefs and practices are constitutive of each other. During the interviews, the actors were encouraged to present their own experiences and develop arguments based on their own activities. Data were collected by drawing on the situated histories and personal positions of those involved: intensive fieldwork was necessary to ensure ‘situational analysis’, embracing the actor’s perspectives seriously by focusing on both activities and meanings and narratives [34].

To ensure consistency in the perspectives offered by the respondents, the questions were based on their participation in the task force and on developing and implementing new targeted public action instruments [35], such as hotel shelters or new individual alarms and police follow-ups.

### 2.3. Data Analysis

The information gathered through these interviews, in addition to the data collection already underway, allowed for an increasingly comprehensive and nuanced view of the discourses and discussions on the issue of intimate partner violence since the beginning of the COVID-19 health crisis and the development of public intervention. Data analysis was organized within our research group, which has conducted extensive fieldwork on IPV in the frame of a former research project (BR/175/A4/IPV-PRO&POL). A reflexive thematic analysis of each interview was performed [36] with a three-step coding (initial; focused; and theoretical coding), by adopting an approach based on “thematic entry points” in line with the theoretical framework of multiple streams analysis [24] in agenda-setting and policy development. This coding allowed us to construct a final template comprising themes and subthemes pertaining to the framing of IPV as a policy issue and the mobilization of possible instruments to address the problem during the pandemic. This analysis identified the narratives of specific actors (NGO members, policymakers, and civil servants) and their positions on the issue of mobilization of new resources and development of new policy instruments in the IPV field. Data were collected until a point of data saturation was reached. Saturation was ensured through coupling the use of multiple methods (triangulation) with a saturation grid method and feedback from field actors [37]. Data triangulation helped us unearth traces of hidden constraints and entrenched power relations when framing new policy interventions. The analysis was also applied to other empirical material gathered (within the media analysis but also during the former IPV Pro&Pol project) and discussed with the research team and accompanying committee until a stable story was built [38].

## 3. Results

As our initial data collection has shown, the adoption of health measures in Belgium to contain the COVID-19 pandemic coincided with the rise of the issue of intimate partner violence on the political and media agenda. The spread of COVID-19 can be seen as a ‘game changer for public governance’ [39]. The pandemic is presented as an unpredictable and uncertain problem with no “off of the shelf” solution. To address such problems, the public sector must adapt by building networks and partnerships with the private sector and civil society to better ensure flexible adaptation and create innovative solutions and pragmatic redirection of resources. However, the pandemic cannot be seen as the only factor that enabled this phenomenon. As the data collected becomes richer and more nuanced, we see a more complex accumulation of different factors. In the light of the stream logics developed by Kingdon, we will grasp the multitude of forces that have enabled this change in public policy, giving attention to the three streams in agenda setting. While no single stream can claim to have such an influence on the policy agenda, the conjuncture of several of them can: “major changes emerge when these streams come together” [24]. 

### 3.1. The Pandemic as a Focusing Event 

“*We were confined. No more going in, no more going out, no more weekends, no more leaving, no more arriving. It was brutal and immediate*.”(Extract from an interview with a member of the NGO sector: Table 1 respondent 2, our translation.) On 18 March 2020, the Belgian authorities introduced the first health measures to deal with the COVID-19 pandemic. A lockdown was imposed on the entire population and with it, the various services dealing with domestic violence. This unpredictable and “focusing event” [24] put the issue of IPV at the heart of the concerns. It gave a real boost to the problem stream as described by Kingdon and thus represented an initial influence, a first step towards setting the issue on the political agenda.

However, the phenomenon was not restricted to Belgium alone. A potential increase in domestic violence was causing concern at an international level. Calls for vigilance regarding these situations and more generally regarding violence against women were made by supranational bodies [40]. The latter then directed the attention of national medias and policy makers towards domestic violence. This is an essential element. It legitimized the concerns of the NGOs. Its members remember being able to rely on these statements: “*International institutions have reacted strongly to this situation, whether at the level of the United Nations, the European Parliament or the WHO. There have been institutions that have sounded the alarm, saying: “women, women, women.” And that has been very supportive for feminist discourse*.” (Extract from an interview with a member of the NGO sector: Table 1 respondent 2, our translation.) If there was so much concern and consensus, it was also because of what was actually happening in the countries first affected by the pandemic. The phenomenon was presented as a “shadow pandemic” concurrent with the COVID-19 pandemic. In March 2020, when the lockdown was introduced in Belgium, the authorities were worried that the same situation would develop in the country: “*The public authorities turned to us. They turned to us and said: In Spain, in Italy, this is a phenomenon that is being confirmed and so we are going to face a surge in situations of domestic violence, and we are going to announce that the helpline is a service*.” (Extract from an interview with a member of the NGO sector: Table 1 respondent 4, our translation). The warnings and the effective increase in calls for help [41,42], in countries that had already introduced lockdown were helping to put the issue in the media and on the political agenda in Belgium.

Indeed, the number of calls received by the helplines in Belgium increased. During the very first weeks of the lockdown, the “Écoute Violence conjugale” listeners (IPV dedicated call line) were faced with a considerable increase in calls. They even tripled [43]. Setting up a third line quickly proved necessary: “*We created a 3rd listening line in a few days. We also created a chat to welcome requests* via *keyboards*.” (Extract from an interview with a member of the NGO sector: Table 1, respondent 5, our translation.) The increase in calls represented a tangible element which confirmed the apprehensions. The facts were widely reported in the media. However, the increase of calls should not automatically be interpreted as an increase in cases of violence. Indeed, respondents testify to the wide variety of calls received at the time: “*There was indeed an explosion of calls during the lockdown, but the calls that were on the increase were mainly from relatives of family situations, perpetrators or victims who were worried*. […] *The line was really the receptacle of social and real anguish concerning all these women who were confined with their aggressors or their companions*.” (Extract from an interview with a member of the NGO sector: Table 1, respondent 1, our translation.) They estimate that “*almost a third of the calls were from relatives of victims, who were concerned about the lockdown. It was quite new that the line was used by these people, composed of relatives, friends, colleagues…”* [44]. The link between number of calls and number of IPV cases is still under investigation. Nevertheless, the indicator of number of total calls was largely mediatized. The change in “an indicator recognized as reliable” [24] also contributed to the awareness of the problems and thus reinforced the rise of the issue on the agenda, particularly when the sharp increase of “call numbers” was picked up by the media.

Lockdown and its consequences, potential or real, on IPV thus directed the attention of the public and decision-makers to this issue: “*It has made domestic violence much more tangible in people’s consciousness than before. It used to be an intellectual level and then it moved to an emotional level. It went from the head to the heart*.” (Extract from an interview with a member of the NGO sector: Table 1, respondent 5, our translation.) This focusing event has put the issue on the political agenda. This violence was already present and regularly denounced by the NGOs before the COVID-19 pandemic: now it could no longer be ignored, as it developed as another pandemic. Once the phenomenon came to the fore, there was no longer any question of putting off the necessary responses: “*It also reveals a social problem that was necessarily present, but perhaps less visible. It is its intensification that was revealed at the time of the health crisis. And finally, we did everything we could to provide the necessary resources at that time. It’s difficult to say… Perhaps they would have been there anyway, but with more difficulty? But it was undoubtedly easier to get people to accept it in that context and therefore to maintain it*.” (Extract from an interview with a member of the Cabinet: Table 2, respondent 2, our translation.)

Finally, the extraordinary nature of the events of March 2020 helped raise the issue of IPV on the political agenda, but the problem stream alone cannot explain this phenomenon [24]. It is necessary to consider the articulation of the different streams, relating to problems, solutions, and politics, to explain this rise on the political agenda. The pandemic and the lockdown of March 2020 then appear as a moment of convergence of the different streams, where the attention on the issue of IPV accelerates and takes on a whole new dimension. In the following paragraphs, we will identify the elements that were already in place before the start of the pandemic, and which also contributed to the rise of this issue on the agenda.

### 3.2. A Feminist Political Universe

While the pandemic can be seen as an accelerating factor in bringing the issue of IPV to the forefront, the political context in which it occurs also has a role to play. The new regional governments in 2019 had given the responsibilities of cabinets for equality to feminist female ministers. All French-speaking governments were convinced of the need to develop common actions to actively tackle the violence against women within a gender based frame. The changes in the political landscape, which took place shortly before the pandemic must therefore be taken into account when considering the phenomenon of agenda-setting.

The NGO sector is aware of this peculiar legislature with a strong feminist stance. They themselves recognize its influence on the rise of the issue of violence against women, and more particularly IPV, on the political agenda. When asked about the management of the issue at the beginning of the pandemic, they chose to dwell on the context: *“Firstly, we were lucky to have only women in the ministries concerned. This is also a conjunction of the stars that has never existed before. This is one of the first chances. And these women have decided to work together*.” (Extract from an interview with a member of the NGO sector: Table 1 respondent 1, our translation). If the commitment of these women politicians to the fight against violence against women was essential, its translation into teamwork was also important for the sector. It shows the importance of this gathering behind shared objectives with common frames at a time when the COVID-19 pandemic was being faced: “*We must recognize that this legislature corresponds to an ideal. We have political and public leaders, regardless of which party is in power, who are all women involved in inequality issues, really involved*. […] *They have clearly decided to put the general interest of the issue above values or party quarrels, and this is reflected in a real political will. This is unique outside the health crisis. We felt it and we feel it very strongly. This positioning has probably made it possible to manage the crisis*.” (Extract from an interview with a member of the NGO sector: Table 1, respondent 2, our translation). At the level of the French-speaking federated entities, this management was organized within a “Task Force on Domestic Violence”. This task force, which was set up in March 2020, also embodies the desire of the French-speaking ministers to work together effectively on issues of violence against women. It brings together the different levels of French-speaking authorities the relevant cabinets, and their administration, but also invites representatives of civil society to the table. (Members of the French-speaking Task Force: Representatives of Linard, Glatiny, Trachte Maron, Morreale, Ben Hamou. Representatives of the Equal and Equal Opportunities administrations. Representatives of associations: Action sociale, CPVCF, Solidarité Femmes, ONE, AMA, ARCA, CVFE, Pôles de ressources.) We will come back to this organization at various points later in this article.

Next, the special character of this legislature does not stop at the French-speaking federated entities. Indeed, whether the government that took office in October 2020 or the one that preceded it, the presence of women in federal governing bodies also influences the political agenda: “*There was a new federal government with a cabinet at federal level in equalitý chance that was obviously more proactive than the previous one, but a Prime Minister who was less favorable than the previous one. That’s it, that’s for the temporalitý at the inter-federal level*.” (Extract from an interview with a member of the Cabinet: Table 2, respondent 1, our translation). The imposition of the issue of IPV initiated by French-speaking politicians was supported and confirmed when the federal Secretary of State for Gender Equality, Equal Opportunities and Diversity, Sarah Schlitz, took office. In November 2020, at the beginning of her term of office, she soon adopted the ‘Federal Action Plan to Combat Gender and Domestic Violence following the 2nd wave of COVID-19’. This “emergency plan” (Extract from an interview with a member of the Cabinet: Table 2 respondent 8, our translation) materialized as the Secretary of State’s commitment to combat violence against women in the context of the pandemic. More broadly, this mandate represented a new element confirming the evolution of the political stream initiated by the French-speaking women politicians. This was a choice of the presidents of the political parties who oversee choosing their ministers and specifying their lines of actions. Yet the government in Flanders, the largest Dutch-speaking region in the country, did not have a Minister in charge of gender issues or women’s rights.

### 3.3. Toward a Common Gender-Based Framing of IPV 

A third trend that contributes to the political agenda of a problem is the “adoption of solutions shared by specialists in the sector” [24]. This sector, as far as the treatment of domestic violence in Belgium is concerned, does not have a common, stabilized frame of reference. It is “seen […] rather as a mosaic of frameworks in tension, not very clear to those seeking help nor to analysts of the sector” [45]. However, the last decade have seen notable developments in this area. Particularly, the Istanbul Convention (Council of Europe, 2011) appeared to be a key moment for the actors in the sector. Ratified in 2016 by Belgium, it is the “first legally binding instrument to combat violence against women and domestic violence. It aims to help European countries act in four major areas: the prevention of violence, protection of victims, prosecution of perpetrators, and the development of integrated, comprehensive, and coordinated policies” (NAP 2015–2019). Adopting such a tool has had a strong impact in the stream of solutions in the sense that Belgium is supposed to embrace a gender-based reading of violence.

The ‘National Action Plan to combat gender-based violence 2021–2025’ is a good sign of the strong commitment of the federal secretary of state Sarah Schlitz: she presents this as a more ambitious plan than her predecessors’, thought out and negotiated with all governments: “*This had never happened before*. […] *The federal government is finally taking on its coordinating role, and this was necessary. The associations on the ground have been asking for this for years. Moreover, in our country, with this decentralized dynamic, sitting around the table and adopting an interfederal plan is very significant. It shows that there is a feminist momentum on the issue of violence, which has become a real topic for politicians, thanks to the work of feminist associations*.” [46] The publication of this NAP can thus be seen as a confirmation of the rise of the issue of violence against women on the political agenda thanks to the construction of a common action plan to address the problem. At the same time, it imposes a coordination between the actions of all involved authorities, for example, between the justice and police federal departments, as well as the health sector and regional actors. Indeed, “*at the moment, the issue of violence is a bit of a hot topic because all these plans that have been published over the last few years did not exist or barely existed. It’s a fairly recent thing. Governments are finally taking the measure of the problem and with the health crisis, there are colossal means that did not exist either*.” (Extract from an interview with a member of the NGO sector: Table 1 respondent 4, our translation).

This latest development within the solutions stream is already having a lasting impact on the violence sector, as the NAP 2021–2025 bears the marks of this. The NAP 2021–2025 is indeed more ambitious than its predecessors, mobilizing more projects for action, more coordination, and more resources. It certainly reflects the Secretary of State’s commitment to violence against women, but also recent developments and the challenge of adopting a common frame of reference: *“The previous NAP dated from 2015, before the signing of the Istanbul Convention: it was purely administrative and did not include measurable objectives… We had the advantage of the ratification of the Convention. But not all parties had the same level of understanding, so we had to make them aware of our international obligations*” [46].

## 4. Discussion

### 4.1. March 2020 as a Window of Opportunity

The issue of violence against women and, more specifically, domestic violence, came to the forefront of the media and political agenda during the COVID-19 pandemic. In recent years, through the various streams that have just been described, the issue has been gaining ground. Although one of these pressure tactics is not enough to really set the issue on the political agenda, the convergence of these pressure tactics accelerated the phenomenon. In March 2020, we witnessed a tangle of streams, the opening of “windows of opportunity” [24]. These are stealthy opportunities during which these different streams can be coupled, generating political changes which are translated into new policy interventions. These take the form of ‘problem windows’ and ‘political windows’. When the streams that push the issue of violence onto the agenda converge, these windows simultaneously open wide, offering new possibilities for action to the various actors involved.

In this context, it is first and foremost a political window that is opening. As we have seen, the presence of feminist ministers in the French-speaking federated entities corresponds to “*a conjunction of the stars that has never existed before*.” (Extract from an interview with a member of the NGO sector: Table 1 respondent 1, our translation). Not necessarily linked to the policy stream or the problem stream, an electoral change may be the reason for the opening of this window of opportunity. In terms of political support, the «*special legislature»* (Extract from an interview with a member of the NGO sector: Table 1 respondent 2, our translation) coming into place in 2019 really embodies this phenomenon and developed a common frame as the UN denounced the risk of a second pandemic in IPV. These women politicians united around “*very clear convergences* […] *in the field of women’s rights and the fight against violence against women”* (Extract from an interview with a member of the Cabinet: Table 2 respondent 3, our translation) embody a window of opportunity to address these issues throughout their mandate. They were reinforced in October 2020 by Sarah Schlitz as Federal Secretary of State for Gender Equality, Equal Opportunities and Diversity, which confirmed this “*feminist Momentum on the issue of violence”* [46].

If the issues of violence start to rise on the political agenda in this way, things will be accelerated: “*What has happened has been a kind of accelerator and an upheaval in relation to public policies that were already being built, not only with us, to be honest.”* (Extract from an interview with a member of the Cabinet: Table 2 respondent 4, our translation). The pandemic and the sanitary measures applied to deal with it opened a second window of opportunity. Lockdown highlighted violence against women, in particular domestic violence, on an international scale. It is now entering the newly opened window of opportunity and is making its way onto the political agenda. “The sudden emergence of problems can lead to the reactivation of existing solutions” [24]. Thus, concerns and demands made over the past few years come to the table: “*The context was such that no one could ignore the fact that it was necessary to have adequate staff to receive these women in an emergency, to welcome them and react correctly, to make the statement, to find a room, to alert the magistrate, etc. It was obvious in addressing the issue that there was of the lack of adequate staff and a lack of adequate resources. It was a no-brainer to have these discussions. This would have been more difficult three years earlier*.” (Extract from an interview with a member of the administrations: Table 2 respondent 7, our translation). To develop new actions, policy entrepreneurs try to use the opportunity to rapidly propose lines of policy actions as solutions to the problems in the spotlight. The major issue is to rapidly design solutions which are consistent with the emerging public vision of the problem. They must also fit with the main political discourses of the policymakers in charge of the issue. NGOs and policymakers worked closely together to design a common line of policy actions.

### 4.2. Women Policy Entrepreneurs

The first political entrepreneurs, that can be identified as such, the ones who instinctively come to mind, are namely the women politicians mentioned above: Benedicte Linard, Christie Morreale, Nawal Ben Hamou and Sarah Schlitz. By looking at the political stream as it allowed an issue to rise the agenda, it became clear that it was less a question of the positioning of a party or coalition, but more of the common strategy of political personalities. These women chose “*to put the general interest of the issue above party lines or party quarrels*.” (Extract from an interview with a member of the NGO sector: Table 1, respondent 5, our translation). This is a notable element that must be considered “*unique* […] *outside of the health crisis.”* (Extract from an interview with a member of the NGO sector: Table 1, respondent 3, our translation). As soon as they took office, these women politicians announced that they wanted to make the issue of violence against women a priority during their mandate. Their common will materialized at the end of 2019 through the Interministerial Conference on Women’s Rights which was developed within a gender perspective. Their initiative bears witness to their influence on the political agenda and the way in which they are helping to push the issue of violence against women up the federal agenda. In March 2020, when the problem stream was aligned with the political stream that was already underway, the issue of IPV became a real topic.

As the issue of domestic violence is increasingly highlighted, also internationally, their right to speak out is strengthened. It is no longer a question of putting forward measures specific to a political program, but of dealing with a situation whose urgency is recognized beyond the national territory. At the heart of the pandemic, their right to speak becomes a necessity for action. The political and social connections inherent in the figure of the political entrepreneur also benefit from the exceptional nature of the pandemic. Indeed, if it can be assumed that these women politicians already had a well-developed network before March 2020, then this network only became stronger. The task force played an important role here. The mechanism stands out because it has enabled the creation and strengthening of links and exchanges between the actors concerned by the issue: “*Now we know almost all the staff in the offices of the five French-speaking ministers. It’s true that it made the task easier. We are recognizable. They recognize us. They know who we are, who we represent. It’s true that it made access and contacts easier*.” (Extract from an interview with a member of the NGO sector: Table 1 respondent 4, our translation). Their network is then considerably growing. This is an element that stands out and which is even more valuable when discussing the relationships between the actors from the sector. Indeed, the creation and maintenance of these relationships during each legislature can sometimes be frustrating: “*Each time, we must start again. Sometimes they don’t even know who we are. We, the specialized services, have been here for 35 years. And we must present ourselves as young people who are just starting out*. […] *We had to start all over again, even though we had built links with each other. We had a partnership, an exchange and everything was thrown on the floor and we had to start again from scratch*.” (Extract from an interview with a member of the NGO sector: Table 1 respondent 5, our translation). If we are discussing the impact on these relationships from the point of view of political figures, it can also be from the point of view of the NGO sector. The role of the latter in placing the issue of domestic violence on the political agenda is also considerable. This is discussed in the following paragraphs.

### 4.3. NGO Actors as Policy Entrepreneurs

While the people mentioned above can easily be seen as political entrepreneurs, the role played by NGO members concerned with the issue throughout the pandemic must be highlighted as well.

The task force created at the very beginning of the pandemic was required to “*monitor the situation, the reception and support infrastructures*, […] *in order to identify the needs and emergencies encountered and to provide a rapid and effective response*” (Extract from an interview with a member of the administrations: Table 2, respondent 7, our translation). The NGO sector is at the heart of the mechanism, and they present themselves as spokespersons of the field actors. In concrete terms, the task force is organized around them: “*The agenda has not changed since the first day, it is the same. In the same order. We always start by giving the floor to the “Domestic Violence Hotline” to see if there is an increase in calls and how they see things on the ground. And then, afterwards, we give the floor to the shelters*.*”* (Extract from an interview with a member of the administrations: Table 2, respondent 7, our translation). As the issue rises on the political agenda, representatives of the NGO sector become indispensable. Their resources as political entrepreneurs and their influence on the agenda are not only valued, but also grow considerably.

First, the pandemic and the task-force mechanism have had an impact on the voice of the representatives from the NGO sector. Beyond being the first to be heard in the task force’s working groups, they really influence the decisions that are taken there. Furthermore, their expertise was already recognized before March 2020. Indeed, “*when you work in an administration or in a cabinet, you are not in the field. The barometer is the voluntary sector, it’s the shelters, it’s the domestic violence hotline.*” (Extract from an interview with a member of the administrations: Table 2 respondent 5, our translation). However, it is used in a different way within the framework of this system. The exceptional and urgent nature of the context at the time of establishing the first lockdown required the continuous collection of data from the field. Not only does this mean that contacts must become more regular, but the nature of the relationship has also evolved into “*consultation and trust from the public authorities regarding the reality on the ground. They are not there and need our feedback. That too creates something special, this proximity with the public authorities.”* (Extract from an interview with a member of the NGO sector: Table 1, respondent 4, our translation). From a relationship based on consultation, a real partnership was established between members of the cabinets, administrations, and representatives of the NGO sector: “*There is confusion between consultation and partnership, and here I find that we are moving more towards partnerships. This is an important step forward. […] Because consultation and cooperation have nothing to do with each other*.” (Extract from an interview with a member of the NGO sector: Table 1 respondent 5, our translation).

Second, the political and social connections of these political entrepreneurs were transformed during COVID. While the representatives of the NGO sector had accumulated contacts over the years and developed quite a tight network, this network has developed further. It represents an even richer resource to be mobilized to put the issue of domestic violence on the political agenda. In connection with the collaborations mentioned above, many of the actors knew each other when they joined the task force. This is an element that facilitated the setting up of the task force: “*It’s in our DNA anyway. We have always worked with the NGO sector. It was the natural reflex*.” (Extract from an interview with a member of the NGO sector: Table 1, respondent 5, our translation). However, this is not the case for all of them. The mechanism set up to deal with a potential increase in violence in the pandemic context allows very concrete contacts: “*The aim of the Task Force was also to get to know each other better and to know, for example, which actor to call upon when faced with a problem. As we got to know each other, I also got to know the field […] of violence*.” (Extract from an interview with a member of the Cabinet: Table 2 respondent 2, our translation.) Beyond the basic creation of a link, the nature of the relationship is also influenced and loses its formality: “*We start to get to know each other. It’s not very formal in the sense that there are no ministers* […] *So there’s this very informal side, based on trust, which allows us to speak freely*.” (Extract from an interview with a member of the Cabinet: Table 2 respondent 1, our translation.) In the end, it is the quality of the relationships that is transformed. Furthermore, the extent of the influence emanating from the establishment and the deepening of these connections can be measured when we think of the members of the NGO sector not represented in the task force. The latter simply could not benefit from their network: *“I was lucky enough to be part of the Task Force, but I wondered: when you were together for months and years now, you know each other so well and you have a little communitý, an entre-soi and not everyone is there.* […] *I measure how lucky I was being solicited and finally, it was us who were doing the feedback from the field based on our own realities but was it the realitý of all and did our needs correspond to the needs of all?*” (Extract from an interview with a member of the NGO sector: Table 1, respondent 7, our translation.) This experience offers them unprecedented “*direct access to the cabinets*” and contributes to make them political entrepreneurs.

Finally, and this is reflected in the preceding paragraphs, the representatives of the NGO sector have an additional resource for influencing the political agenda: their years of experience. Indeed, they have been around for a long time and have become key structures in the domestic violence sector, as they are “*the specialized services that have been here for 35 years.”* (Extract from an interview with a member of the NGO sector: Table 1, respondent 2, our translation.) The frustration expressed earlier illustrates the different time frames in which the practices and associations in the field operate. Within the task force, time was also a resource that could be mobilized by representatives of the NGO sector in addition to the right to speak as expert to sympathetic ears. Enjoying political recognition enables them to become political entrepreneurs and thus influence the political agenda: “*The fact that we are heard that our opinion is being considered. Our opinion changes things, changes things. That’s new. That’s new*. […] *It’s not just about hearing. I think it goes even further: it’s co-construction*.” (Extract from an interview with a member of the NGO sector: Table 1, respondent 4, our translation).

## 5. Conclusions

“The access to the agenda […] depends on the fortuitous meeting of streams and the existence of political entrepreneurs who sometimes succeed in coupling these streams in an active way” [24]. As we saw it, March 2020 was a special moment. The pandemic and the lockdown of the population brought domestic violence to light and to the forefront of the political agenda. Different types of policy entrepreneurs have managed to take advantage of this specific situation. Indeed, as we have seen, women ministers are not the only ones to have acted during this period. Actors of the NGOs have also participated in creating opportunities in this pandemic. To face the pandemic, the “task force” has been a particular space allowing for the creation of new dynamics between actors from different sectors. It was a space for meeting and for co-creation. This group is described as the “COVID’S lesson” by its members. All actors (policymakers, experts, and NGOs) decided to engage in policy developments. How was this developed? How did the “entrepreneurs” play the game to ensure their success? The analysis points to the impact of the pandemic as a moment of policy redefinition: policy issues do not come as established problems but are undergoing a whole process of redefinition and debates to arrive at a common vision in terms of causalities and target groups. This framing is an ongoing effort for policymakers to put the issue on the agenda and to try to maintain it. Thanks to Kingdon’s model, we have been able to better understand this phenomenon in the Belgian context and to address it in its complexity. We have appreciated the international dimension in media agenda-setting and its impact in the development on the task force taking IPV with a gender approach.

However, attention must be paid to the solutions that have been implemented in this specific context and which received the TF support. As we have seen, some of them had already been thought of before the pandemic. Some others already existed and have been reinforced: for example, new resources were made available for increasing “online support” and this service was expanded and further professionalized. Some solutions have also been inspired by what is usually done in other sectors. As an example, temporary hotel accommodation, as we have seen opened in Brussels, had already been used before to offer shelter to homeless people. These are solutions created in a world without pandemics or without stringent health measures: how can we translate them during COVID-19? We can then ask ourselves: what needs do these solutions meet? Do they cover the needs of IPV victims? Are they adapted to the capacities of the professionals helping them? This paper has addressed the exceptional agenda-setting dynamics of domestic violence during the COVID-19 pandemic in the French-speaking part of Belgium. The streams model was relevant for this purpose, but the model does not consider the time horizon. When the focusing event is as sudden, and political support provides immediate unexpected resources, one tends to reach for readymade solutions. Future research could further investigate the impact of such actions taken during this period on (and from) the practices of professionals and the capacity of the specialists to maintain a long-term action program.

Finally, what is happening now, more than two years after the first lockdown? The public can also turn to other problems when other sectors of policy intervention receive more media attention and generate increasing public outcry. Fighting violence requires complex interventions and sustained policy attention, while policymakers (and the media as well) are confronted with a multitude of legitimate policy problems that are competing for their attention [26]. While the NGOs succeeded in developing active and efficient advocacy work and took advantage of the political opportunities during the pandemics, other issues important to the public and the policymakers gained more attention on the governmental agenda: new economic problems due to a sluggish recovery, and the recent energy crisis due to the war in Ukraine. Feminist actors must stabilize the results of the efforts and policy developments which were achieved during the pandemic. Through the different plans, written during or after the pandemic, they are already trying to ensure structural changes to make sure that the gains of this phenomenon are not lost.

## Figures and Tables

**Table 1 ijerph-20-04461-t001:** Summary table of interviews with civil society.

	Respondent	Gender
1	Representative of the French-speaking hotline domestic violence	Female
2	Representative the French-speaking shelter federation	Female
3	Representative of Association of Aid to perpetrators	Female
4	Representative of the Collective against family violence and exclusion	Male
5	Representative of a shelter for women victims of violence	Female
6	Representative of Association of Pharmacists in March 2020	Male
7	Representative of the Office of birth and childhood.	Female
8	Representative of an « SOS Enfants » team	Female
9	Representative of the Department of Public Prosecutions of Liège	Female

**Table 2 ijerph-20-04461-t002:** Summary table of interviews with the members of administrations and political cabinets.

	Respondent	Gender	Territory of Action	Competences
1	Member of Trachte-Marron ministerial Cabinet	Female	Région Bruxelles-Capitale	Health Promotion, Family, Budget and Public Service. Economic transition and scientific research.
2	Member of Linard ministerial Cabinet.	Female	Fédération Wallonie-Bruxelles	Childhood, Health, Women’s rights, culture, media
3	Member of Morreale ministerial Cabinet.	Female	Région Wallonne	Employment, Formation, Health, Women’s rights, Equal Opportunities
4	Member of Ben Hamou ministerial Cabinet.	Female	Region Bruxelles-Capitale	Housing, Equal Opportunities.
5	Member of Institute for Gender Equality	Male	Belgium	Federal public institution that promotes equality between women and men.
6	Member of Glatiny ministerial Cabinet.	Female	Région Wallonne	Youth welfare, Justice centres.
7	Member of the Institute for Equal Opportunities	Female	Fédération Wallonie-Bruxelles	Institution of the Fédération Wallonie-Bruxelles that promotes equal opportunities
8	Member of the Schlitz ministerial Cabinet.	Female	Belgium	Equal Opportunities, Gender equality and Diversity.

**Table 3 ijerph-20-04461-t003:** Core questions of the interview grid.

	Themes Covered:	Example of Questions Used:
*To start the conversation*	Identity of the person, its experiences, and its overview of the IPV sector during the pandemic.	- Could you introduce yourself, your (previous) function? - According to you, how has the pandemic impacted domestic violence and the IPV sector? - …
*Follow-up questions*	- First moments of the crisis and adaptations of the sector. - Interactions between actors of IPV sector during the pandemic. - Its participation in the Task Force.	- How was the announcement of the lockdown received in your association/federation/cabinet? - What has been decided at that moment/ what decisions/actions have been taken? - Who were you in contact with? - Have you been warned about the potential issues of the lockdown? How and by whom? - What were your first needs at that moment and how did you reach it? - Could you tell me more about the “Task force violence conjugale et intrafamiliale”? How did it start? How was it organized? Who was part of it? And how would you sum up your experience in this group? - What kind of action has been taken by the Task force? Could you explain these actions in detail? - Who did you collaborate with? - What other actions have been taken as the pandemic progressed? - Now, in hindsight, what do you think about this period? - …
*To end the conversation*	- Its look back on the pandemic and its effect on the IPV sector	- What lesson have you learnt from this period? - According to you, is there something important we didn’t discuss? - …

## Data Availability

The data presented in this study are available on request from the corresponding author. The data are not publicly available due to GDPR Provision.

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
