# Peer review of "How Did the COVID-19 Pandemic Increase Salience of Intimate Partner Violence on the Policy Agenda?"

_ijerph, 2023, doi:10.3390/ijerph20054461_

Round 1

Reviewer 1 Report (Previous Reviewer 3)

Dear Luce Lebrun,

Congratulations, you improved manuscript quality.

I suggest minor changes to accept it.

Line 3: Please, use this structure for your names and affiliation: “Firstname Lastname 1

Line 6 to 18: According to template form, use a structured abstract.

References: Footnote references are not adequate for this journal, please, include this information like us general references.

Line 49 and 72: Don’t use a special tabulation in these lines.

Annexes: This information must be placed in an independent document. You entitled it like Figure but is a Table.

According to template form, you must include this information:

Supplementary Materials: The following supporting information can be downloaded at: www.mdpi.com/xxx/s1, Figure S1: title; Table S1: title; Video S1: title.

Author Contributions: For research articles with several authors, a short paragraph specifying their individual contributions must be provided. The following statements should be used “Conceptualization, X.X. and Y.Y.; methodology, X.X.; software, X.X.; validation, X.X., Y.Y. and Z.Z.; formal analysis, X.X.; investigation, X.X.; resources, X.X.; data curation, X.X.; writing—original draft preparation, X.X.; writing—review and editing, X.X.; visualization, X.X.; supervision, X.X.; project administration, X.X.; funding acquisition, Y.Y. All authors have read and agreed to the published version of the manuscript.” Please turn to the CRediT taxonomy for the term explanation. Authorship must be limited to those who have contributed substantially to the work reported.

Funding: Please add: “This research received no external funding” or “This research was funded by NAME OF FUNDER, grant number XXX” and “The APC was funded by XXX”. Check carefully that the details given are accurate and use the standard spelling of funding agency names at https://search.crossref.org/funding. Any errors may affect your future funding.

Institutional Review Board Statement: In this section, you should add the Institutional Review Board Statement and approval number, if relevant to your study. You might choose to exclude this statement if the study did not require ethical approval. Please note that the Editorial Office might ask you for further information. Please add “The study was conducted in accordance with the Declaration of Helsinki, and approved by the Institutional Review Board (or Ethics Committee) of NAME OF INSTITUTE (protocol code XXX and date of approval).” for studies involving humans. OR “The animal study protocol was approved by the Institutional Review Board (or Ethics Committee) of NAME OF INSTITUTE (protocol code XXX and date of approval).” for studies involving animals. OR “Ethical review and approval were waived for this study due to REASON (please provide a detailed justification).” OR “Not applicable” for studies not involving humans or animals.

Informed Consent Statement: Any research article describing a study involving humans should contain this statement. Please add “Informed consent was obtained from all subjects involved in the study.” OR “Patient consent was waived due to REASON (please provide a detailed justification).” OR “Not applicable.” for studies not involving humans. You might also choose to exclude this statement if the study did not involve humans.

Written informed consent for publication must be obtained from participating patients who can be identified (including by the patients themselves). Please state “Written informed consent has been obtained from the patient(s) to publish this paper” if applicable.

Data Availability Statement: We encourage all authors of articles published in MDPI journals to share their research data. In this section, please provide details regarding where data supporting reported results can be found, including links to publicly archived datasets analyzed or generated during the study. Where no new data were created, or where data is unavailable due to privacy or ethical restrictions, a statement is still required. Suggested Data Availability Statements are available in section “MDPI Research Data Policies” at https://www.mdpi.com/ethics.

Acknowledgments: In this section, you can acknowledge any support given which is not covered by the author contribution or funding sections. This may include administrative and technical support, or donations in kind (e.g., materials used for experiments).

Conflicts of Interest: Declare conflicts of interest or state “The authors declare no conflict of interest.” Authors must identify and declare any personal circumstances or interest that may be perceived as inappropriately influencing the representation or interpretation of reported research results. Any role of the funders in the design of the study; in the collection, analyses or interpretation of data; in the writing of the manuscript; or in the decision to publish the results must be declared in this section. If there is no role, please state “The funders had no role in the design of the study; in the collection, analyses, or interpretation of data; in the writing of the manuscript; or in the decision to publish the results”.

Author Response

Dear Reviewer,

Thank you for your follow-up and for your encouragement.

We’ve seen that your last comments and recommendations were more about the form and information missing for the submission. We have taken into account these changes in the last version and will taking it into account submitting the paper. Since this is our first submission to this journal, your advices were very helpful. We also had one more reviewing by a native speaker, Loes Diricks, to improve the language. Please see the attachment.

Thanks for the time spend on this paper.

Kind regards,

Luce Lebrun

Reviewer 2 Report (New Reviewer)

The discussion needs to be sharpened by providing references, both those that support and those that do not support it so that the analysis is more accurate

There are some references that are very old (eg 1963, 2008 etc.), maybe the author needs to search more carefully and find references similar to the latest edition.

Detailed input can be seen in the review file

Author Response

Dear Reviewer,

Thank you for your follow-up. Here you will find the point-to-point response to the comment you made concerning this fourth reviewing step.

Point 1 :  “It is better to write down the goals in a transparent, clear and firm manner” Response 1 : The goals of the paper are described in the section "Aim of the study". For more clarity, we have decided to number them according to the "discussion section"

Point 2 : “There are some references that are very old (eg 1963, 2008 etc.), maybe the author needs to search more carefully and find references similar to the latest edition.”  Response 2 : Thanks to your comment, we have reviewed our references.The 1963 reference has been removed and replaced by a more recent reference.Here is the list of the references that have been updated/added in this last version :

  • Thiry, A. & Fallon, C. (2023). Repenser la prise en charge des IPV : d’une logique de judiciarisation à un dynamique d’apprentissage. In Vanneste C. et al. (Eds), Regards croisés sur la Violences entre partenaires intimes, Politeia
  • Guba, E. G., & Lincoln, Y. S. (2005). Paradigmatic Controversies, Contradictions, and Emerging Confluences. In: Denzin, N.K. and Lincoln, Y.S., Eds., The Sage Handbook of Qualitative Research, 3rd Edition, Sage, Thousand Oaks, 191-215..
  • McCombs, M., Shaw, D., Weaver, D. (2014). New Directions in Agenda-Setting Theory and Research. Mass Communication & Society. 17 (6): 781–802. doi:1080/15205436.2014.964871.
  • Kingdon, J. W. (1995) Agendas, Alternatives and Public Policy. (2nd ed.). New York: Pearson.

Point 3 : “The discussion needs to be sharpened by providing references, both those that support and those that do not support it so that the analysis is more accurate”      Response 3 : We do not really understand this comment. The aim of the paper is to analyse the data collected with the Kingdon's model. It is through this theoretical framework that we analyze the political context and the placing on the agenda of the issues related to domestic violence.The interviews made are really the heart of this paper. But that could be the starting point of another paper on this issue.

Point 4 : “It is better if the conclusion is the final statement of the research based on the results of the analysis. In conclusion, there should be no need for further references and do not repeat the discussion”                                                                                                                                                                        Response 4 : Thanks for your comment. The changes made in the last version are highlighted (in yellow) in the attachment. Thanks to your comment, we also paid more attention to not have new information or reference in the conclusion.

Finally, we also had the paper re-read a second time by a native speaker. In the attachment, you will be able to see all the improvements made by Loes Diricks, our translator, to improve the language.

Thanks for the time spend on this paper.

King regards, Luce Lebrun

Reviewer 3 Report (New Reviewer)

Conclusively, I suggest accepting this article after a minor to moderate revision. The reason is missing some essential information. This critical information is listed below:
1. This article seems to present qualitative research. The authors asked respondents some questions and had some discussions. Section 3 of this manuscript presented these discussions. Nevertheless, readers of this article may think that Section 3 is irrelevant to those questions shown in Figure 1. I suggest the authors revise their manuscript by adding the sentence "From the question......, we have the discussions.......".
2. The authors stated that this article presented some interviews with some members of the NGO sector. Please revise the relevant footnotes by adding the names of those members or other information identifying them. Are they listed in Table 1 or 2? If the answer is not, I suggest the authors revise these two tables.

Author Response

Dear reviewer,

Thank you for your follow-up and encouragement. Here you will find the point-to-point response.

Point 1 : “This article seems to present qualitative research. The authors asked respondents some questions and had some discussions. Section 3 of this manuscript presented these discussions. Nevertheless, readers of this article may think that Section 3 is irrelevant to those questions shown in Figure 1. I suggest the authors revise their manuscript by adding the sentence "From the question......, we have the discussion”

Response 1 : As it is explained in the method section (line 202), the data were collected in an abductive approach. As such, the purpose of exploratory interviews was not to verify hypotheses, but rather to become aware of dimensions and aspects of a problem that the researcher would probably not have thought of spontaneously. Then, the interview grid is not supposed to be seen as an exhaustive list of all the question asked during the interview. It rather gathers all the different topic/theme that have been discussed. As you said, it could be misleading for the readers. In order to prevent this, we have reviewed our interview grid by highlight the theme that correspond to the questions. You will find the updates in the annexes attachment.

Point 2 : “The authors stated that this article presented some interviews with some members of the NGO sector. Please revise the relevant footnotes by adding the names of those members or other information identifying them. Are they listed in Table 1 or 2? If the answer is not, I suggest the authors revise these two tables.”

Response 2 : The extract of the interviews are the ones listed in the table 1&2. However, our research protocol does not allow us to give the name of the people we met. We have then anonymized the interview on purpose. However, your comment was right and in order to be clearer for the reader, we update our footnotes being more precise about the which interview we are referencing to. We will find these improvements in the attachment.

Finally, we also had the paper re-read a second time by a native speaker. In the attachment, you will be able to see all the improvements made by Loes Diricks, our translator.

Thanks for the time spend on this paper.

King regards,Luce Lebrun

Round 2

Reviewer 2 Report (New Reviewer)

Almost all the references are too old and not up to date. We recommend that you add the latest references (at least the last 3 years) and from reputable journals.

It would be better if this online reference was included in the reference

Author Response

Dear reviewer, 

Thanks for your follow-up. In this new submission, some recent references have been added. The online references have also been included in the reference, as recommended. 

Please see the attachment with the modifications in track change. 

Kind regards, 

Luce Lebrun

This manuscript is a resubmission of an earlier submission. The following is a list of the peer review reports and author responses from that submission.

Round 1

Reviewer 1 Report

The article makes good use of Kingdon's streams theory to analyze the agenda setting opportunity provided by the Covid-19 restrictions for the issue of domestic violence in Belgium. It critically engages effectively with the model, both using it to good effect and exposing its limitations. It highlights the important issue of what activists are to do once the media focus moves on. 

I think it is ready for publication, I just have one small change (a deletion of 'the' in the conclusion) to make it read smoothly. 

Author Response

Dear reviewer,

Thank you for the time spend on this paper, for your advice and encouragements. It really helped us to go deeply in the reviewing process. However, we made some modifications according to the other reviewers’ comments. If you want to read this new version,  please see the attachment.

Thank again for helping me to improve this paper.

Sincerely,

Luce Lebrun

Reviewer 2 Report

From reading the revised manuscript, it was clear to me that the authors had taken some of my comments into account and I appreciate that! For example, the abstract was clearer, and the language had improved.

More importantly, however, I felt a little unsatisfied with how the authors had dealt with some of my other comments.
As part of revising this paper a second time, I first had a quick look through the response letter and my immediate impression was that the authors had made major revisions in response to my comments. I then started to read the revised introduction. I noted down some issues (e.g., I found the introduction to be unclear and disconnected from the paper itself).

Before I moved on to reviewing the method section, I went back to the authors' response where I noted that I had actually raised these issues earlier, but that the authors had not responded to them.

As I then read on, I noted down additional issues, and as I went back to the response letter, I noted that I had raised these issues earlier. Again, I noted that the authors had listed my comment in their response to me, but not addressed it specifically nor made revisions to the text. I understand that this may occasionally happen in a revision process and normally I would not have a problem with pointing this out in a second revision round. However, as this occurred on more than a couple of occasions, I did not feel compelled to continue reviewing the results and discussion.

For the reasons listed above, I recommend either rejecting the manuscript for publication or inviting the authors to revise the manuscript again, but to engage with ALL the comments that were raised and refer to page and line numbers to guide me to where they have made revisions to avoid any potential misunderstandings.

Author Response

Dear reviewer,

Thank for this second reviewing round. We were a bit disappointed to read that the improvements made to the first submission were not consistent enough. In the first reviewing round, we were grateful to read all your constructive feedbacks. It has indeed motivated us to deeply go into the reviewing process. We sadly discover that the second submission has simply made you stop reading before the result section. In this third submission, we then made some improvements according to your new comments but also tried to better highlight the ones made before. Since this is my first paper submission, I’m not used to this process, and I certainly made mistakes. I will try being clearer in this point-to-point response, by using the pages and lines number to guide you as you recalled me to do so.

ABSTRACT + Key words:

We were glad to read that you found the abstract and the language clearer. We have indeed completely written this section and make the entire paper re-read by a native speaker. In this third submission, we made very small changes (a few word) and add some key words (page 1, line 19).

INTRODUCTION:

In this section, I will response to comments made in the first and second reviewing process as you had the feeling that we did not took into account the issues your raised at the very beginning of the reviewing process.

Point 1: “Structure the introduction differently: pandemic, political, and media agenda, and only then, clearly and separately, the study objectives and research questions. This will avoid confusion between pre-existing literature and the purpose of the paper.”

Response 1:  We completely changed the structure of the introduction. The section talking about the literature and the particular context of the paper (section 1.1.1 & section 1.1.2) are then starting on page 2 (line 49). Only then, we have the new section about the aim of the paper and the concepts (section 1.2), starting on page 3 (line 136).

Point 2: “I found the introduction to be unclear and disconnected to the paper itself”

Response 2: We hope the changes made in the structure contribute to make the introduction clearer. We also add a small paragraph (page 4, line 158) in order to smooth the transition towards the “materials and method” section and avoid a “disconnected introduction”.

METHOD:

In this section, we will refer to comments made in the first reviewing process. Indeed, the comments made in the second round were about the fact that improvements made were not consistent enough.

Point 1 : “The ID number”

Response 1 : Concerning this point, we do not have a protocol number for this part of the research, which concerns a qualitative analysis of media discourse and political-administrative actors during the covid crisis. The ethics committee of the University of Liège was consulted for the other parts of the research (interviews with victims and perpetrators of domestic violence). The research project has been validated and is funded by the Belgian Federal Science Policy. You can find more information about the research project (in which the data collection used in this paper is from) here : https://www.belspo.be/belspo/organisation/about_en.stm

Point 2: “The semi-structured interview questions”.

Response 2: We add an “annexes section” where you can find the semi-structured questions (p.14). However, we also tried to be clearer about our qualitative method. We will find the additions we made on page 6 (line 209-229).

Point 3 : “A summary table of demographic information”

Response 3: We add two table with demographic information in order to be clearer about the people we met during these interviews: - Table 1. Summary table of interviews with the civil society. (p.5, line204) - Table 2. Summary table of interviews with the members of administrations and political cabinets. (p.5, line 206).

Point 4: “Data analysis is completely missing”

Response 4: A “data analysis” section was added to the paper. It was indeed completely missing in the first submission. We will find this new section on page 6-7 (line 230-251).

RESULTS + DISCUSSION + CONCLUSION:

For these sections, since the second submission made you stop reading after the method section, we will try to highlight the improvements made to the first and second submission. As you will see, we completely reorganised the paper according to the comments of our different reviewers. We find the result section (page 7, line 252), a discussion section (page 10, line 425) and a conclusion (page 13, line 571). This way, we hope that the added value of this paper is highlighted. Please, see the attachement. We also add some elements in order to better explain the belgian political context (Page 8 (line 339) + page 9 (line 379)).

Thank again for helping me to improve this paper.

Sincerely,

Luce Lebrun

Reviewer 3 Report

Dear authors,

Your manuscript is not adequate to be published, in my opinion don't have quality enough.

Manuscript haven’t MDPI format, please use the Microsoft Word template or LaTeX template.

Some lines included a font size bigger without reason.

References must be numbered in order of appearance in the text and listed individually at the end of the manuscript.

In the text, reference numbers should be placed in square brackets [ ], and placed before the punctuation; for example [1], [1–3] or [1,3]. The reference list should include the full title, as recommended by the ACS style guide.

Introduction include little information to context this subject.

Results and Discussion include interesting information, but it's neccesary a restructuring.

Please, reevaluate your manuscript.

Author Response

Dear reviewer,

Thank you for the time spend on this paper, for your advice and encouragements. It really helped us to go deeply in the reviewing process.

Point 1 : template

Response 1: We will find this new submission in the adequate template.

Point 2: References.

Response 2: We have rewritten our references according to your advices. I hope it is now appropriate.

Point 3 : Structure

Response 3 : In order to clarify our argument, we re-structure the entire paper : The section talking about the literature and the particular context of the paper (section 1.1.1 & section 1.1.2) are then starting on page 2 (line 49). Only then, we have the new section about the aim of the paper and the concepts (section 1.2), starting on page 3 (line 136). Then we hope that the result, discussion and conclusion are clearer.

Qualitative method : We also add some clarification about our data and method : We add an “annexes section” where you can find the semi-structured questions (p.14). However, we also tried to be clearer about our qualitative method. We will find the additions we made on page 6 (line 209-229).We add two table with demographic information in order to be clearer about the people we met during these interviews: - Table 1. Summary table of interviews with the civil society. (p.5, line204) - Table 2. Summary table of interviews with the members of administrations and political cabinets. (p.5, line 206). A “data analysis” section was added to the paper. It was indeed completely missing in the first submission. We will find this new section on page 6-7 (line 230-251).

We also made some small modifications according to the other reviewers’ comments. If you want to read this new version,  please see the attachment.

Thank again for helping me to improve this paper.

Sincerely,

Luce Lebrun
